# Design and Synthesis of Novel Podophyllotoxins Hybrids and the Effects of Different Functional Groups on Cytotoxicity

**DOI:** 10.3390/molecules27010220

**Published:** 2021-12-30

**Authors:** Zhongtao Yang, Zitong Zhou, Xiai Luo, Xiaoling Luo, Hui Luo, Lianxiang Luo, Weiguang Yang

**Affiliations:** 1The Marine Biomedical Research Institute, Guangdong Medical University, Zhanjiang 524023, China; yangzt@gdmu.edu.cn (Z.Y.); zzt15766229745@163.com (Z.Z.); luoxiai83@163.com (X.L.); 2The Marine Biomedical Research Institute of Guangdong Zhanjiang, Zhanjiang 524023, China; 3School of Basic Medical Sciences, Guangxi Medical University, Nanning 530021, China; luoxiaoling@gxmu.edu.cn

**Keywords:** podophyllotoxin hybrids, CuAAC, antiproliferative activity, molecular docking, molecular dynamics, structure–activity correlation

## Abstract

Development of novel anticancer therapeutic candidates is one of the key challenges in medicinal chemistry. Podophyllotoxin and its derivatives, as a potent cytotoxic agent, have been at the center of extensive chemical amendment and pharmacological investigation. Herein, a new series of podophyllotoxin-*N*-sulfonyl amidine hybrids (**4a**–**4v**, **5a**–**5f**) were synthesized by a CuAAC/ring-opening procedure. All the synthesized podophyllotoxins derivatives were evaluated for in vitro cytotoxic activity against a panel of human lung (A-549) cancer cell lines. Different substituents’, or functional groups’ antiproliferative activities were discussed. The –CF_3_ group performed best (IC_50_: 1.65 μM) and exhibited more potent activity than etoposide. Furthermore, molecular docking and dynamics studies were also conducted for active compounds and the results were in good agreement with the observed IC_50_ values.

## 1. Introduction

Podophyllotoxin is a potent cytotoxic agent and serves as a useful lead compound for the development of drugs due to its cytotoxic, insecticidal, antifungal, antiviral, anti-inflammatory, neurotoxic, immunosuppressive, antirheumatic, antioxidative, anti-spasmogenic, and hypolipidemic activities. Its derivatives also act as excellent anticancer aspirants for future chemotherapies [1,2,3,4,5]. Similar to etoposide [6], teneposide [7] drugs have been proven to be potential antitumor drugs in clinical use; NPF [8] and GL-331 [9] have been under clinical trials. However, their therapeutic efficacy is often limited to their undesirable secondary effects, for example, gastrointestinal toxicity, neurotoxicity, hair loss, bone marrow suppression, etc., as well as the development of resistance by cancer cells [3]. Therefore, the development of less toxic derivatives or analogues has been a consistent focus of podophyllotoxin modifications.

Previous studies have shown that modifications in the podophyllotoxin skeleton’s A-ring and D-ring compounds appeared to be biologically less active than podophyllotoxin itself [10]. Intact A-ring and D-ring systems are important for the compounds’ DNA topoisomerase II (dtop II)-inhibiting activity. Meanwhile, rational modification at C4, including podophyllotoxin hybrids [11,12,13] and the β-configurated derivative podophyllotoxin hybrids, might improve the molecule’s topoisomerase II-inhibition activity, water solubility, drug resistance profile, and antimitotic activities [14] (Figure 1).

Depending on the chosen modern molecule hybridization strategy [15,16], podophyllotoxin/epipodophyllotoxin can hybridize with amino acid/peptide [17,18], azole [19], biotin [20], carbamate [21], chalcone [22], cinnamic acid [23], cinnamic acid [24], ferrocene [25], furan/pyrrole [26], hydroxamic acid [27], indole [28], norcantharidin [29], pyridine/pyrimidine [30], pterostilbene [31], retinoic acid [32], sugar [33], (thio)urea [34], or sulfur [35] (Figure 1). These have been subjected to extensive research and have been shown to be highly active against both MDR and drug-sensitive cancer cell lines with IC_50_ or GI_50_ values at a nanomolar level; however, the water solubility, bioactivities, and toxicity still need to be improved. Analogs on a large scale, for drug discovery and application, are still required.

On the other hand, the *N*-sulfonyl amidine moiety frequently appears in biologically active compounds, and has displayed a crucial role in medicinal chemistry research, where it has therapeutic effects against several diseases. It showed a prolific set of biological activities (Figure 2), including anticancer (**a**) [36], antifungal (**b**) [37], antiresorptive (**c**) [38], dopamine transporter inhibition (**d**) [39], and antiproliferative activities [40], etc.

Accordingly, herein we describe a new series of podophyllotoxin-*N*-sulfonyl amidine hybrids (PSAH) with potent antiproliferative activity. The operationally simple synthesis protocol involved stirring a mixture of the terminal alkynes, sulfonyl azides, and podophyllotoxin in the presence of a copper (I) catalyst; additionally, we studied the effects of different functional groups on cytotoxicity (Figure 1).

## 2. Materials and Methods

### 2.1. General Methods

All melting points were determined on a Yanaco melting point apparatus (Kyoto, Japan) and were uncorrected. IR spectra were recorded as KBr pellets on a Nicolet FT-IR 5DX spectrometer (Waltham, MA, USA). All spectra of ^1^H NMR (400 MHz) and ^13^C NMR (100 MHz) were recorded on a Bruker AVANCE NEO 400 MHz spectrometer (Berne, Switzerland) in DMSO-*d_6_* or CDCl_3_ (unless otherwise indicated), with TMS used as an internal reference and the J values given in Hz. HRMS were obtained on a Thermo Scientific Q Exactive Focus Orbitrap LC-MS/MS spectrometer (Waltham, MA, USA). Optical rotations are measured on a P-2000, serial number: B209161232, JASCO corporation (Tokyo, Japan). 4*β*-Aminopodophyllotoxin (**1a**) and 4′-*O*-demethyl-4*β*-aminopodophyllotoxin (**1b**) were prepared by using reported methods [41,42]. All terminal alkynes (**2a**–**2m**, see Appendix A) were prepared by the manufacturer and sulfonyl azides (**3a**–**3j**, see Appendix A) were prepared using methods in the literature [43].

### 2.2. General Procedure for the Synthesis of PSAH

We added 4*β*-Aminopodophyllotoxin (**1a**) or 4′-*O*-demethyl-4*β*-aminopodophyllotoxin (**1b**) (0.1 mmol), terminal alkynes (**2**) (0.12 mmol), sulfonyl azides (**3**) (0.12 mmol), CuI (0.02 mmol), NEt_3_ (0.4 mmol), and dry MeCN (2 mL) to an oven-dried Schlenk tube, equipped with a magnetic stirring bar. After the reaction was stirred at room temperature for 12 h, the solvent was removed by evaporation in vacuum. The residue was directly purified by flash column chromatography (silica gel, using hexanes/EtOAc as eluent) to afford the corresponding products PSAH, see **4** or **5**. Details of the compound characterizations are provided in the following subsections.

*N-((5S,5aS,8aR,9R)-8-Oxo-9-(3,4,5-trimethoxyphenyl)-5,5a,6,8,8a,9-hexahydrofuro[3′,4′:6,7]naphtho[2,3-d][1,3]dioxol-5-yl)-2-phenyl-N′-tosylacetimidamide* (**4a**), white solid, 63 mg; yield: 92%; m.p.: 146–148 °C, [α]_D_^25^ = −59.1 (c 1.00, CH_2_Cl_2_); ^1^H NMR (400 MHz, CDCl3) δ = 7.82 (dd, J = 15.7, 7.9 Hz, 2H), 7.33 (dd, J = 11.9, 7.4 Hz, 5H), 7.21 (d, *J* = 7.2 Hz, 2H), 6.52 (s, 1H), 6.41 (s, 1H), 6.17 (s, 2H), 5.95 (d, *J* = 3.0 Hz, 2H), 5.27 (d, *J* = 6.9 Hz, 1H), 5.19 (t, *J* = 5.7 Hz, 1H), 4.42 (d, *J* = 5.1 Hz, 1H), 4.41–4.27 (m, 2H), 4.10 (t, *J* = 8.3 Hz, 1H), 3.77 (s, 3H), 3.71 (s, 6H), 3.61 (t, *J* = 10.0 Hz, 1H), 2.86–2.68 (m, 1H), 2.45 (m, 1H), 2.43 (s, 3H); ^13^C NMR (100 MHz, CDCl_3_) *δ* = 173.9, 166.5, 152.8 (2C), 148.8, 147.9, 143.2, 140.0, 137.5, 134.5, 133.0, 132.6, 129.9, 129.8 (2C), 129.7 (2C), 128.6, 127.3, 126.6 (2C), 126.5, 110.3, 108.7, 108.4 (2C), 101.9, 69.0, 60.9, 56.4 (2C), 50.6, 43.7, 42.0, 39.5, 36.9, 21.7; IR *ν*_max_ (KBr): 3398, 2943, 1778, 1535, 1485, 1423, 1392, 1234, 933, 690 cm^−1^; HRMS (ESITOF) *m/z* calcd. for C_37_H_36_N_2_O_9_S, [M + H]^+^ 685.2214, found 685.2209.

*N-((5S,5aS,8aR,9R)-8-Oxo-9-(3,4,5-trimethoxyphenyl)-5,5a,6,8,8a,9-hexahydrofuro[3′,4′:6,7]naphtho[2,3-d][1,3]dioxol-5-yl)-2-(p-tolyl)-N′-tosylacetimidamide* (**4b**), white solid, 65 mg; yield: 93%; m.p.: 148–150 °C, [α]_D_^25^ = −69.1 (c 1.00, CH_2_Cl_2_); ^1^H NMR (400 MHz, CDCl_3_) *δ* = 7.84 (d, *J* = 8.1 Hz, 2H), 7.31 (d, *J* = 8.0 Hz, 2H), 7.16–7.04 (m, 4H), 6.53 (s, 1H), 6.42 (s, 1H), 6.18 (s, 2H), 5.96 (s, 2H), 5.33 (d, *J* = 7.0 Hz, 1H), 5.20 (dd, *J* = 7.0, 4.7 Hz, 1H), 4.43 (d, *J* = 5.0 Hz, 1H), 4.29 (m, 2H), 4.10 (dd, *J* = 9.2, 7.4 Hz, 1H), 3.77 (s, 3H), 3.71 (s, 6H), 2.79 (dddd, *J* = 15.3, 11.5, 7.4, 4.7 Hz, 1H), 2.45 (m, 1H), 2.43 (s, 3H), 2.32 (s, 3H); ^13^C NMR (100 MHz, CDCl_3_) *δ* = 173.9, 166.9, 152.8 (2C), 148.8, 147.9, 143.2, 140.0, 138.4, 137.5, 134.5, 132.6, 130.5 (2C), 129.7 (3C), 129.6 (2C), 127.4, 126.6 (2C), 110.3, 108.8, 108.4 (2C), 101.8, 69.1, 60.9, 56.4 (2C), 50.5, 43.7, 42.0, 39.1, 36.9, 21.7, 21.3; IR *ν*_max_ (KBr): 3394, 2927, 1778, 1531, 1485, 1388, 1330, 1234, 933, 690 cm^−^^1^; HRMS (ESITOF) *m/z* calcd. for C_38_H_38_N_2_O_9_S, [M + H]^+^ 699.2371, found 699.2364.

*2-(4-(Dimethylamino)phenyl)-N-((5S,5aS,8aR,9R)-8-oxo-9-(3,4,5-trimethoxyphenyl)-5,5a,6,8,8a,9-hexahydrofuro[3′,4′:6,7]naphtho[2,3-d][1,3]dioxol-5-yl)-N′-tosylacetimidamide* (**4c**), white solid, 70 mg; yield: 96%; m.p.: 139–141 °C, [α]_D_^25^ = −77.3 (c 1.00, CH_2_Cl_2_); ^1^H NMR (400 MHz, CDCl_3_) *δ* = 7.84 (d, *J* = 8.0 Hz, 2H), 7.30 (d, *J* = 8.0 Hz, 2H), 7.01 (d, *J* = 8.4 Hz, 2H), 6.64 (d, *J* = 8.4 Hz, 2H), 6.57 (s, 1H), 6.41 (s, 1H), 6.19 (s, 2H), 5.94 (dd, *J* = 5.3, 1.3 Hz, 2H), 5.54 (d, *J* = 7.1 Hz, 1H), 5.23 (dd, *J* = 7.2, 4.8 Hz, 1H), 4.43 (d, *J* = 5.0 Hz, 1H), 4.20 (d, *J* = 4.4 Hz, 2H), 4.16–4.10 (m, 1H), 3.77 (s, 3H), 3.71 (s, 6H), 3.62 (dd, *J* = 10.9, 9.3 Hz, 1H), 2.93 (s, 6H), 2.80 (dddd, *J* = 15.1, 11.3, 7.1, 4.6 Hz, 1H), 2.47 (dd, *J* = 14.5, 5.1 Hz, 1H), 2.42 (s, 3H); ^13^C NMR (100 MHz, CDCl_3_) *δ* = 174.0, 167.9, 152.7 (2C), 150.2, 148.6, 147.8, 142.9, 140.1, 137.4, 134.5, 132.5, 130.6 (2C), 129.6 (2C), 127.4, 126.5 (2C), 119.3, 113.2 (2C), 110.2, 108.8, 108.3 (2C), 101.7, 69.0, 60.8, 56.3 (2C), 50.3, 43.6, 41.9, 40.4 (2C), 38.5, 36.8, 21.6; IR *ν*_max_ (KBr): 3437, 2931, 1778, 1527, 1485, 1284, 1234, 1145, 941, 686 cm^−1^; HRMS (ESITOF) *m/z* calcd. for C_39_H_41_N_3_O_9_S, [M + H]^+^ 728.2636, found 728.2632.

*2-(4-Methoxyphenyl)-N-((5S,5aS,8aR,9R)-8-oxo-9-(3,4,5-trimethoxyphenyl)-5,5a,6,8,8a,9-hexahydrofuro[3′,4′:6,7]naphtho[2,3-d][1,3]dioxol-5-yl)-N′-tosylacetimidamide* (**4d**), white solid, 68 mg; yield: 95%; m.p.: 133–135 °C, [α]_D_^25^ = −91.3 (c 1.00, CH_2_Cl_2_); ^1^H NMR (400 MHz, CDCl_3_) *δ* = 7.76 (t, *J* = 5.7 Hz, 2H), 7.28 (m, 2H), 7.12 (d, *J* = 8.2 Hz, 2H), 6.87–6.75 (m, 2H), 6.53 (d, *J* = 2.3 Hz, 1H), 6.37 (d, *J* = 6.6 Hz, 1H), 6.18 (s, 2H), 5.92 (d, *J* = 6.7 Hz, 2H), 5.84 (m, 1H), 5.21 (dd, *J* = 7.1, 4.7 Hz, 1H), 4.40 (d, *J* = 5.1 Hz, 1H), 4.28 (dd, *J* = 16.8, 4.1 Hz, 1H), 4.19–4.00 (m, 2H), 3.76 (s, 3H), 3.74 (s, 3H), 3.69 (s, 6H), 3.60 (t, *J* = 10.1 Hz, 1H), 2.79 (dddd, *J* = 15.0, 11.3, 7.3, 4.7 Hz, 1H), 2.60 (dd, *J* = 14.1, 5.2 Hz, 1H), 2.41 (s, 3H); ^13^C NMR (100 MHz, CDCl_3_) *δ* = 174.0, 167.0, 159.3, 152.6 (2C), 148.6, 147.7, 143.0, 139.9, 137.2, 134.6, 132.5, 130.8, 130.7, 129.6, 129.5, 127.3, 126.4, 126.3, 124.8, 114.9, 114.8, 110.1, 108.7, 108.2 (2C), 101.7, 68.9, 60.7, 56.3 (2C), 55.3, 50.4, 43.6, 41.8, 38.4, 36.8, 21.5; IR *ν*_max_ (KBr): 3390, 2935, 2839, 1778, 1585, 1512, 1330, 1238, 933, 690 cm^−^^1^; HRMS (ESITOF) *m/z* calcd. for C_38_H_38_N_2_O_10_S, [M + H]^+^ 715.2320, found 715.2321.

*2-(3-Hydroxyphenyl)-N-((5S,5aS,8aR,9R)-8-oxo-9-(3,4,5-trimethoxyphenyl)-5,5a,6,8,8a,9-hexahydrofuro[3′,4′:6,7]naphtho[2,3-d][1,3]dioxol-5-yl)-N′-tosylacetimidamide* (**4e**), white solid, 55 mg; yield: 79%; m.p.: 154–156 °C, [α]_D_^25^ = −48.1 (c 1.00, CH_2_Cl_2_). dr = 3.3:1; ^1^H NMR (400 MHz, CDCl_3_, contains two isomers) *δ* = 7.73 (d, *J* = 8.0 Hz, 2H, major), 7.65 (d, *J* = 7.9 Hz, 2H, minor), 7.30–7.23 (m, 2H, major + minor), 7.18 (t, *J* = 7.8 Hz, 1H, minor), 7.09 (t, *J* = 8.0 Hz, 1H, major), 6.84 (dd, *J* = 8.2, 2.3 Hz, 2H, minor), 6.72 (d, *J* = 7.0 Hz, 2H, major), 6.68 (d, *J* = 7.5 Hz, 1H, major + minor), 6.50 (s, 1H, major), 6.39 (s, 1H, minor), 6.26 (s, 1H, minor), 6.23 (s, 1H, major), 6.21 (d, *J* = 6.8 Hz, 2H, minor), 6.16 (s, 2H, major), 5.85 (d, *J* = 14.7 Hz, 2H, major), 5.79 (d, *J* = 8.5 Hz, 2H, minor), 5.30 (dd, *J* = 8.5, 5.1 Hz, 1H, minor), 5.18 (dd, *J* = 7.3, 3.5 Hz, 1H, major), 4.32 (d, *J* = 3.9 Hz, 1H, major), 4.24 (d, *J* = 16.0 Hz, 1H, major), 4.17 (d, *J* = 8.5 Hz, 1H, minor), 4.13 (d, *J* = 13.9 Hz, 1H, minor), 4.05–3.97 (m, 2H, major), 3.90–3.84 (m, 2H, minor), 3.80 (s, 3H, minor), 3.73 (s, 3H, major), 3.69 (s, 6H, major + minor), 3.62 (m, 2H, major), 3.38 (dd, *J* = 10.3, 4.6 Hz, 1H, minor), 3.20 (dt, *J* = 10.3, 5.2 Hz, 1H, minor), 2.75 (m, 2H, major + minor), 2.40 (s, 3H, major + minor); ^13^C NMR (100 MHz, CDCl_3_, contains two isomers) *δ* (major + minor) = 179.1, 174.6, 167.1, 166.7, 157.4, 157.1, 153.5 (2C), 152.6 (2C), 148.5, 148.0, 147.6, 147.3, 143.2, 143.1, 139.7, 139.6, 137.2, 137.0, 136.8, 134.8, 134.8, 133.9, 132.5, 131.1, 130.7, 130.5, 129.6 (2C), 127.2, 127.0, 126.4 (2C), 126.2 (2C), 121.5, 120.9, 116.8, 116.6, 116.0, 115.3, 110.1, 109.9, 108.9 (2C), 108.3 (2C), 106.5, 105.0, 101.7, 101.5, 69.1, 68.6, 60.9, 60.8, 56.3 (2C), 56.2 (2C), 50.6, 45.1, 44.7, 43.5, 41.7, 39.6, 39.1, 38.3, 36.9, 21.6 (2C); IR *ν*_max_ (KBr): 3630, 3387, 2927, 1778, 1535, 1485, 1330, 1234, 933, 690 cm^−1^; HRMS (ESITOF) *m/z* calcd. for C_37_H_36_N_2_O_10_S, [M + H]^+^ 701.2163, found 701.2159.

*2-(4-Chlorophenyl)-N-((5S,5aS,8aR,9R)-8-oxo-9-(3,4,5-trimethoxyphenyl)-5,5a,6,8,8a,9-hexahydrofuro[3′,4′:6,7]naphtho[2,3-d][1,3]dioxol-5-yl)-N′-tosylacetimidamide* (**4f**), white solid, 63 mg; yield: 88%; m.p.: 138–140 °C, [α]_D_^25^ = −61.3 (c 1.00, CH_2_Cl_2_); ^1^H NMR (400 MHz, CDCl_3_) *δ* = 7.73 (d, *J* = 7.9 Hz, 2H), 7.28 (m, 4H), 7.16 (d, *J* = 8.1 Hz, 2H), 6.51 (s, 1H), 6.40 (s, 1H), 6.18 (s, 2H), 5.95 (d, *J* = 5.1 Hz, 2H), 5.68 (d, *J* = 6.9 Hz, 1H), 5.19 (dd, *J* = 7.0, 4.7 Hz, 1H), 4.45–4.33 (m, 2H), 4.12 (d, *J* = 16.4 Hz, 1H), 4.05 (t, *J* = 8.3 Hz, 1H), 3.75 (s, 3H), 3.70 (s, 6H), 3.60 (t, *J* = 10.0 Hz, 1H), 2.80 (dddd, *J* = 15.2, 11.6, 7.4, 4.8 Hz, 1H), 2.61 (dd, *J* = 14.4, 5.0 Hz, 1H), 2.42 (s, 3H); ^13^C NMR (100 MHz, CDCl_3_) *δ* = 173.9, 165.8, 152.7 (2C), 148.8, 147.9, 143.3, 139.7, 137.4, 134.5, 134.3, 132.6, 131.7, 130.9 (2C), 129.7 (2C), 129.6 (2C), 127.3, 126.4 (2C), 110.3, 108.7, 108.3 (2C), 101.8, 68.9, 60.8, 56.4 (2C), 50.7, 43.7, 41.9, 38.6, 36.8, 21.6; IR *ν*_max_ (KBr): 3302, 2927, 1778, 1535, 1485, 1234, 933, 810, 779, 690 cm^−1^; HRMS (ESITOF) *m/z* calcd. for C_37_H_35_ClN_2_O_9_S, [M + H]^+^ 719.1824, found 719.1821.

*2-(3-Chlorophenyl)-N-((5S,5aS,8aR,9R)-8-oxo-9-(3,4,5-trimethoxyphenyl)-5,5a,6,8,8a,9-hexahydrofuro[3′,4′:6,7]naphtho[2,3-d][1,3]dioxol-5-yl)-N′-tosylacetimidamide* (**4g**), white solid, 68 mg; yield: 94%; m.p.: 151–153 °C, [α]_D_^25^ = −59.6 (c 1.00, CH_2_Cl_2_); ^1^H NMR (400 MHz, CDCl_3_) *δ* = 7.65 (d, *J* = 7.9 Hz, 2H), 7.27–7.19 (m, 4H), 7.20–7.10 (m, 2H), 6.50 (s, 1H), 6.33 (s, 1H), 6.27 (d, *J* = 7.0 Hz, 1H), 6.17 (s, 2H), 5.90 (d, *J* = 16.2 Hz, 2H), 5.21 (dd, *J* = 7.1, 4.1 Hz, 1H), 4.38 (dd, *J* = 10.1, 5.6 Hz, 2H), 4.05–3.97 (m, 1H), 3.97 (d, *J* = 11.9 Hz, 1H), 3.72 (s, 3H), 3.68 (s, 6H), 3.58 (t, *J* = 9.5 Hz, 1H), 2.86–2.66 (m, 2H), 2.40 (s, 3H); ^13^C NMR (100 MHz, CDCl_3_) *δ* = 174.1, 165.4, 152.5 (2C), 148.5, 147.6, 143.1, 139.6, 137.1, 135.9, 134.8, 134.7, 132.4, 130.4, 129.5 (2C), 129.2, 127.9, 127.4, 127.3, 126.2 (2C), 110.1, 108.7, 108.1 (2C), 101.6, 68.7, 60.7, 56.2 (2C), 50.5, 43.6, 41.6, 38.3, 36.7, 21.5; IR *ν*_max_ (KBr): 3398, 2931, 1778, 1585, 1423, 1330, 1234, 933, 771, 690 cm^−^^1^; HRMS (ESITOF) *m/z* calcd. for C_37_H_35_ClN_2_O_9_S, [M + H]^+^ 719.1825, found 719.1824.

*2-(4-Bromophenyl)-N-((5S,5aS,8aR,9R)-8-oxo-9-(3,4,5-trimethoxyphenyl)-5,5a,6,8,8a,9-hexahydrofuro[3′,4′:6,7]naphtho[2,3-d][1,3]dioxol-5-yl)-N′-tosylacetimidamide* (**4h**), white solid, 59 mg; yield: 77%; m.p.: 163–165 °C, [α]_D_^25^ = −64.4 (c 1.00, CH_2_Cl_2_); ^1^H NMR (400 MHz, CDCl_3_) *δ* = 7.68 (d, *J* = 8.2 Hz, 2H), 7.40 (d, *J* = 8.3 Hz, 2H), 7.29–7.20 (m, 2H), 7.08 (d, *J* = 8.1 Hz, 2H), 6.51 (s, 1H), 6.37 (s, 1H), 6.17 (s, 2H), 5.97–5.94 (m, 2H), 5.93 (m, 1H), 5.21 (dd, *J* = 7.1, 4.6 Hz, 1H), 4.41 (d, *J* = 4.9 Hz, 1H), 4.33 (d, *J* = 16.2 Hz, 1H), 4.09–3.97 (m, 1H), 3.73 (s, 3H), 3.69 (s, 6H), 3.60 (t, *J* = 9.9 Hz, 1H), 2.80 (dddd, *J* = 15.1, 11.2, 7.2, 4.7 Hz, 1H), 2.68 (dd, *J* = 14.4, 5.0 Hz, 1H), 2.41 (s, 3H); ^13^C NMR (100 MHz, CDCl_3_) *δ* = 174.0, 165.7, 152.7 (2C), 148.7, 147.8, 143.2, 139.7, 137.3, 134.6, 132.6, 132.5 (2C), 132.4, 131.1 (2C), 129.6 (2C), 127.3, 126.4, 126.3, 122.2, 110.2, 108.7, 108.3 (2C), 101.8, 68.9, 60.8, 56.3 (2C), 50.7, 43.6, 41.8, 38.6, 36.8, 21.6; IR *ν*_max_ (KBr): 3437, 2935, 1778, 1535, 1485, 1330, 1234, 933, 694, 551 cm^−1^; HRMS (ESITOF) *m/z* calcd. for C_37_H_35_BrN_2_O_9_S, [M + H]^+^ 763.1319, found 763.1310.

*N-((5S,5aS,8aR,9R)-8-Oxo-9-(3,4,5-trimethoxyphenyl)-5,5a,6,8,8a,9-hexahydrofuro[3′,4′:6,7]naphtho[2,3-d][1,3]dioxol-5-yl)-N′-tosyl-2-(4-(trifluoromethyl)phenyl)acetimidamide* (**4i**), white solid, 73 mg; yield: 97%; m.p.: 201–203 °C, [α]_D_^25^ = −57.2 (c 1.00, CH_2_Cl_2_); ^1^H NMR (400 MHz, CDCl3) δ = 7.74–7.65 (m, 2H), 7.56 (dd, J = 8.2, 3.2 Hz, 2H), 7.35 (d, *J* = 7.9 Hz, 2H), 7.25 (dd, *J* = 8.9, 2.9 Hz, 2H), 6.47 (s, 1H), 6.40 (d, *J* = 2.4 Hz, 1H), 6.18 (s, 2H), 5.93 (d, *J* = 13.1 Hz, 2H), 5.81–5.66 (m, 1H), 5.21 (dd, *J* = 6.8, 4.6 Hz, 1H), 4.51 (dd, *J* = 16.1, 3.0 Hz, 1H), 4.46–4.39 (m, 1H), 4.14 (dd, *J* = 16.2, 5.4 Hz, 1H), 4.06 (t, *J* = 8.3 Hz, 1H), 3.74 (s, 3H), 3.69 (s, 6H), 3.64 (t, *J* = 10.1 Hz, 1H), 2.82 (dddd, *J* = 15.1, 11.4, 7.3, 4.7 Hz, 1H), 2.67 (dt, *J* = 13.8, 5.2 Hz, 1H), 2.40 (s, 3H); ^13^C NMR (100 MHz, CDCl_3_) *δ* = 174.0, 165.2, 152.7 (2C), 148.8, 147.9, 143.4, 139.6, 137.6, 137.4, 134.6, 132.6, 130.4 (q, *J* = 31.1 Hz, 1C), 129.8 (2C), 129.6 (2C), 127.3 (q, *J* = 1.1 Hz, 2C), 126.4, 126.3 (q, *J* = 3.3 Hz, 2C), 123.9 (q, *J* = 272.2 Hz, 1C), 110.3, 108.6, 108.3 (2C), 101.9, 68.9, 60.8, 56.4 (2C), 50.9, 43.7, 42.0, 39.0, 36.8, 21.6; IR *ν*_max_ (KBr): 3437, 2935, 1778, 1535, 1485, 1327, 1234, 1126, 933, 690 cm^−1^; HRMS (ESITOF) *m/z* calcd. for C_38_H_35_F_3_N_2_O_9_S, [M + H]^+^ 753.2088, found 753.2082.

*4-(2-(((5S,5aS,8aR,9R)-8-Oxo-9-(3,4,5-trimethoxyphenyl)-5,5a,6,8,8a,9-hexahydrofuro[3′,4′:6,7]naphtha[2,3-d][1,3]dioxol-5-yl)amino)-2-(tosylimino)ethyl)benzoic acid* (**4j**), white solid, 45 mg; yield: 61%; m.p.: 170–172 °C, [α]_D_^25^ = −61.8 (c 1.00, (CH_3_)_2_CO); ^1^H NMR (400 MHz, (CD_3_)_2_CO) *δ* = 8.00 (d, *J* = 7.4 Hz, 1H), 7.93 (d, *J* = 7.9 Hz, 2H), 7.79 (d, *J* = 7.9 Hz, 2H), 7.52 (d, *J* = 7.9 Hz, 2H), 7.33 (d, *J* = 7.9 Hz, 2H), 6.77 (s, 1H), 6.48 (s, 1H), 6.35 (s, 2H), 5.97 (d, *J* = 7.5 Hz, 2H), 5.45 (dd, *J* = 7.6, 4.0 Hz, 1H), 4.62 (d, *J* = 15.0 Hz, 1H), 4.52 (d, *J* = 4.8 Hz, 1H), 4.26 (d, *J* = 15.0 Hz, 1H), 4.13–4.04 (m, 1H), 3.76 (t, *J* = 9.5 Hz, 1H), 3.66 (s, 6H), 3.65 (s, 4H), 3.61 (s, 1H), 3.16–3.01 (m, 2H), 2.39 (s, 3H); ^13^C NMR (100 MHz, (CD_3_)_2_CO) *δ* = 173.6, 166.5, 165.2, 152.5 (2C), 148.1, 147.2, 142.1, 141.2, 140.5, 137.3, 135.4, 132.8, 129.7 (2C), 129.2, 129.1 (4C), 128.4, 126.2 (2C), 109.5, 109.1, 108.5 (2C), 101.5, 68.4, 59.4, 55.4 (2C), 50.2, 43.6, 41.1, 38.4, 37.0, 20.4; IR *ν*_max_ (KBr): 3433, 2924, 1778, 1724, 1543, 1485, 1330, 1234, 933, 690 cm^−^^1^; HRMS (ESITOF) *m/z* calcd. for C_38_H_36_N_2_O_11_S, [M + H]^+^ 729.2113, found 729.2110.

*N-((5S,5aS,8aR,9R)-8-Oxo-9-(3,4,5-trimethoxyphenyl)-5,5a,6,8,8a,9-hexahydrofuro[3′,4′:6,7]naphtho[2,3-d][1,3]dioxol-5-yl)-N′-tosyloctanimidamide* (**4k**), white solid, 56 mg; yield: 81%; m.p.: 140–142 °C, [α]_D_^25^ = −99.3 (c 1.00, CH_2_Cl_2_); ^1^H NMR (400 MHz, CDCl_3_) *δ* = 7.74 (d, *J* = 7.9 Hz, 2H), 7.28 (d, *J* = 5.5 Hz, 2H), 6.69 (s, 1H), 6.43 (s, 1H), 6.21 (s, 2H), 5.95 (d, *J* = 7.5 Hz, 2H), 5.25 (dd, *J* = 7.1, 4.1 Hz, 1H), 4.46 (d, *J* = 4.3 Hz, 1H), 4.04 (dd, *J* = 9.2, 6.8 Hz, 1H), 3.74 (s, 3H), 3.71 (s, 6H), 3.62 (t, *J* = 9.7 Hz, 1H), 2.87 (td, *J* = 17.0, 15.7, 6.1 Hz, 3H), 2.67 (dt, *J* = 14.3, 8.0 Hz, 1H), 2.41 (s, 3H), 1.59 (t, *J* = 7.5 Hz, 2H), 1.32–1.21 (m, 9H), 0.87 (t, *J* = 6.7 Hz, 3H); ^13^C NMR (100 MHz, CDCl_3_) *δ* = 174.2, 169.0, 152.7 (2C), 148.6, 147.8, 142.9, 140.3, 137.2, 134.8, 132.6, 129.5 (2C), 127.8, 126.4 (2C), 110.2, 109.1, 108.2 (2C), 101.8, 69.0, 60.8, 56.3 (2C), 50.2, 43.7, 41.9, 36.8, 33.9, 31.8, 29.6, 28.9, 28.2, 22.6, 21.6, 14.1; IR *ν*_max_ (KBr): 3441, 2927, 1778, 1531, 1485, 1330, 1234, 1145, 937, 694 cm^−1^; HRMS (ESITOF) *m/z* calcd. for C_37_H_44_N_2_O_9_S, [M + H]^+^ 693.2840, found 693.2836.

*N-((5S,5aS,8aR,9R)-8-Oxo-9-(3,4,5-trimethoxyphenyl)-5,5a,6,8,8a,9-hexahydrofuro[3′,4′:6,7]naphtho[2,3-d][1,3]dioxol-5-yl)-2-(thiophen-2-yl)-N′-tosylacetimidamide* (**4l**), white solid, 42 mg; yield: 61%; m.p.: 128–130 °C, [α]_D_^25^ = −82.9 (c 1.00, CH_2_Cl_2_); ^1^H NMR (400 MHz, CDCl_3_) *δ* = 7.81 (d, *J* = 7.8 Hz, 2H), 7.30 (d, *J* = 8.0 Hz, 3H), 6.97 (d, *J* = 5.6 Hz, 2H), 6.60 (s, 1H), 6.42 (s, 1H), 6.19 (s, 2H), 5.96 (s, 2H), 5.76 (d, *J* = 6.9 Hz, 1H), 5.21 (t, *J* = 5.7 Hz, 1H), 4.63–4.47 (m, 2H), 4.46 (d, *J* = 5.0 Hz, 1H), 4.05 (t, *J* = 8.3 Hz, 1H), 3.76 (s, 3H), 3.71 (s, 6H), 3.62 (t, *J* = 10.1 Hz, 1H), 2.89–2.73 (m, 1H), 2.57 (dd, *J* = 14.3, 5.1 Hz, 1H), 2.42 (s, 3H); ^13^C NMR (100 MHz, CDCl_3_) *δ* = 173.9, 165.2, 152.7 (2C), 148.8, 147.9, 143.3, 139.8, 137.5, 134.5, 133.9, 132.6, 129.8, 129.7, 129.2, 128.0, 127.3, 127.1, 126.6, 126.5 (2C), 110.3, 108.9, 108.3 (2C), 101.8, 69.0, 60.8, 56.4 (2C), 50.6, 43.7, 42.0, 37.0, 33.5, 21.7; IR *ν*_max_ (KBr): 3441, 2935, 1778, 1585, 1539, 1485, 1419, 1392, 933, 690 cm^−1^; HRMS (ESITOF) *m/z* calcd. for C_35_H_34_N_2_O_9_S, [M + H]^+^ 691.1778, found 691.1774.

*3-(1H-Indol-1-yl)-N-((5S,5aS,8aR,9R)-8-oxo-9-(3,4,5-trimethoxyphenyl)-5,5a,6,8,8a,9-hexahydrofuro[3′,4′:6,7]naphtho[2,3-d][1,3]dioxol-5-yl)-N′-tosylpropanimidamide* (**4m**), white solid, 47 mg; yield: 64%; m.p.: 149–151 °C, [α]_D_^25^ = −78.0 (c 1.00, CH_2_Cl_2_); ^1^H NMR (400 MHz, CDCl_3_) *δ* = 7.82 (dd, *J* = 8.1, 3.0 Hz, 2H), 7.65 (d, *J* = 7.8 Hz, 1H), 7.44 (d, *J* = 8.1 Hz, 1H), 7.30 (d, *J* = 8.0 Hz, 2H), 7.17 (t, *J* = 7.9 Hz, 1H), 7.14 (d, *J* = 3.1 Hz, 1H), 7.10 (t, *J* = 7.4 Hz, 1H), 6.50 (d, *J* = 3.1 Hz, 1H), 6.28 (d, *J* = 2.9 Hz, 1H), 6.08 (s, 2H), 5.91 (d, *J* = 4.2 Hz, 2H), 5.88 (s, 1H), 5.67 (d, *J* = 6.8 Hz, 1H), 4.80 (dt, *J* = 23.7, 4.3 Hz, 2H), 4.71–4.60 (m, 1H), 4.24 (d, *J* = 5.1 Hz, 1H), 3.73 (m, 1H), 3.70 (s, 3H), 3.64 (s, 6H), 3.46 (dt, *J* = 12.3, 5.9 Hz, 1H), 3.30–3.15 (m, 2H), 2.55 (dddd, *J* = 15.2, 11.6, 7.3, 3.6 Hz, 1H), 2.41 (s, 3H), 2.18–2.09 (m, 1H); ^13^C NMR (100 MHz, CDCl_3_) *δ* = 174.1, 164.7, 152.5 (2C), 148.5, 147.4, 143.3, 139.8, 137.2, 135.7, 134.8, 132.3, 129.6 (2C), 128.8, 127.8, 127.0, 126.4 (2C), 122.3, 121.9, 120.0, 109.9, 109.2, 109.1, 108.2 (2C), 102.7, 101.6, 68.7, 60.7, 56.3 (2C), 50.7, 44.0, 43.5, 41.4, 36.6, 35.8, 21.6; IR *ν*_max_ (KBr): 3433, 2931, 1778, 1581, 1535, 1508, 1330, 1234, 933, 694 cm^−1^; HRMS (ESITOF) *m/z* calcd. for C_40_H_39_N_3_O_9_S, [M + H]^+^ 738.2480, found 738.2478.

*N-((5S,5aS,8aR,9R)-8-Oxo-9-(3,4,5-trimethoxyphenyl)-5,5a,6,8,8a,9-hexahydrofuro[3′,4′:6,7]naphtho[2,3-d][1,3]dioxol-5-yl)-2-phenyl-N′-(phenylsulfonyl)acetimidamide* (**4n**), white solid, 64 mg; yield: 96%; m.p.: 135–137 °C, [α]_D_^25^ = −71.2 (c 1.00, CH_2_Cl_2_); ^1^H NMR (400 MHz, CDCl_3_) *δ* = 7.98–7.86 (m, 2H), 7.60–7.44 (m, 4H), 7.33–7.29 (m, 2H), 7.25–7.18 (m, 2H), 6.53 (s, 1H), 6.37 (s, 1H), 6.17 (s, 2H), 5.92 (dd, *J* = 6.5, 1.2 Hz, 2H), 5.67 (d, *J* = 7.0 Hz, 1H), 5.21 (dd, *J* = 7.0, 4.7 Hz, 1H), 4.46–4.33 (m, 2H), 4.24 (d, *J* = 16.6 Hz, 1H), 4.07 (td, *J* = 9.2, 7.4 Hz, 1H), 3.75 (s, 3H), 3.69 (s, 6H), 3.64–3.54 (m, 1H), 2.80 (dddd, *J* = 15.3, 11.6, 7.4, 4.7 Hz, 1H), 2.54 (dd, *J* = 14.4, 5.1 Hz, 1H); ^13^C NMR (100 MHz, CDCl_3_) *δ* = 174.0, 166.7, 152.7 (2C), 148.7, 147.8, 142.7, 137.4, 134.6, 133.1, 132.5, 132.4, 129.6 (2C), 129.2, 129.0 (2C), 128.4, 127.3, 126.4 (2C), 126.3, 110.2, 108.8, 108.3 (2C), 101.8, 68.9, 60.8, 56.3 (2C), 50.6, 43.6, 41.8, 39.4, 36.8; IR *ν*_max_ (KBr): 3317, 2939, 1778, 1542, 1419, 1334, 1234, 933, 729, 690 cm^−1^; HRMS (ESITOF) *m/z* calcd. for C_36_H_34_N_2_O_9_S, [M + H]^+^ 671.2058, found 671.2053.

*N′-((2,3-Dihydro-1H-inden-5-yl)sulfonyl)-N-((5S,5aS,8aR,9R)-8-oxo-9-(3,4,5-trimethoxyphenyl)-5,5a,6,8,8a,9-hexahydrofuro[3′,4′:6,7]naphtho[2,3-d][1,3]dioxol-5-yl)-2-phenylacetimidamide* (**4o**), white solid, 69 mg; yield: 97%; m.p.: 164–166 °C, [α]_D_^25^ = −67.8 (c 1.00, CH_2_Cl_2_); ^1^H NMR (400 MHz, CDCl_3_) *δ* = 7.72 (s, 1H), 7.68 (d, *J* = 8.0 Hz, 1H), 7.30 (dd, *J* = 9.1, 7.0 Hz, 4H), 7.21 (d, *J* = 7.1 Hz, 2H), 6.52 (s, 1H), 6.38 (s, 1H), 6.17 (s, 2H), 5.93 (d, *J* = 7.5 Hz, 2H), 5.56 (d, *J* = 6.9 Hz, 1H), 5.25–5.18 (m, 1H), 4.43–4.33 (m, 2H), 4.24 (d, *J* = 16.7 Hz, 1H), 4.09 (t, *J* = 8.3 Hz, 1H), 3.76 (s, 4H), 3.70 (s, 7H), 3.61 (t, *J* = 10.1 Hz, 1H), 2.95 (td, *J* = 7.6, 2.9 Hz, 4H), 2.79 (dddd, *J* = 15.2, 11.6, 7.2, 4.7 Hz, 1H), 2.52 (dd, *J* = 14.4, 5.1 Hz, 1H), 2.12 (p, *J* = 7.5 Hz, 2H); ^13^C NMR (100 MHz, CDCl_3_) *δ* = 174.0, 166.4, 152.7 (2C), 149.5, 148.7, 147.8, 145.3, 140.6, 137.4, 134.6, 133.2, 132.5, 129.6 (2C), 129.5 (2C), 128.3, 127.4, 124.8, 124.7, 122.4, 110.2, 108.8, 108.3 (2C), 101.8, 69.0, 60.8, 56.4 (2C), 50.5, 43.6, 41.9, 39.3, 36.9, 32.9, 32.8, 25.4; IR *ν*_max_ (KBr): 3398, 2939, 2843, 1778, 1535, 1330, 1234, 1130, 933, 698 cm^−1^; HRMS (ESITOF) *m/z* calcd. for C_39_H_38_N_2_O_9_S, [M + H]^+^ 711.2371, found 711.2370.

*N′-((4-Chlorophenyl)sulfonyl)-N-((5S,5aS,8aR,9R)-8-oxo-9-(3,4,5-trimethoxyphenyl)-5,5a,6,8,8a,9-hexahydrofuro[3′,4′:6,7]naphtho[2,3-d][1,3]dioxol-5-yl)-2-phenylacetimidamide* (**4p**), white solid, 62 mg; yield: 87%; m.p.: 230–232 °C, [α]_D_^25^ = −77.0 (c 1.00, CH_2_Cl_2_); ^1^H NMR (400 MHz, CDCl_3_) *δ* = 7.82–7.75 (m, 2H), 7.48–7.40 (m, 2H), 7.31 (d, *J* = 7.1 Hz, 3H), 7.24–7.15 (m, 2H), 6.52 (s, 1H), 6.37 (s, 1H), 6.17 (s, 2H), 5.93 (d, *J* = 4.2 Hz, 2H), 5.87 (d, *J* = 7.1 Hz, 1H), 5.18 (dd, *J* = 7.1, 4.7 Hz, 1H), 4.41 (d, *J* = 5.0 Hz, 1H), 4.39–4.15 (m, 2H), 4.08 (dd, *J* = 9.1, 7.5 Hz, 1H), 3.74 (s, 3H), 3.69 (s, 6H), 3.63 (dd, *J* = 10.9, 9.2 Hz, 1H), 2.81 (dddd, *J* = 15.2, 11.5, 7.3, 4.6 Hz, 1H), 2.58 (dd, *J* = 14.4, 5.0 Hz, 1H); ^13^C NMR (100 MHz, CDCl_3_) *δ* = 173.9, 166.8, 152.7 (2C), 148.7, 147.8, 141.2, 138.7, 137.4, 134.5, 133.0, 132.5, 129.6 (2C), 129.5 (2C), 129.2 (2C), 128.4, 127.9 (2C), 127.1, 110.2, 108.7, 108.3 (2C), 101.8, 68.8, 60.8, 56.3 (2C), 50.6, 43.6, 41.8, 39.4, 36.8; IR *ν*_max_ (KBr): 3390, 2920, 1778, 1535, 1392, 1330, 1234, 929, 756, 636 cm^−1^; HRMS (ESITOF) *m/z* calcd. for C_36_H_33_ClN_2_O_9_S, [M + H]^+^ 705.1668, found 705.1663.

*N′-((4-Bromophenyl)sulfonyl)-N-((5S,5aS,8aR,9R)-8-oxo-9-(3,4,5-trimethoxyphenyl)-5,5a,6,8,8a,9-hexahydrofuro[3′,4′:6,7]naphtho[2,3-d][1,3]dioxol-5-yl)-2-phenylacetimidamide* (**4q**), white solid, 65 mg; yield: 86%; m.p.: 232–234 °C, [α]_D_^25^ = −70.4 (c 1.00, CH_2_Cl_2_); ^1^H NMR (400 MHz, CDCl_3_) *δ* = 7.71 (d, *J* = 8.3 Hz, 2H), 7.60 (d, *J* = 8.2 Hz, 2H), 7.39–7.25 (m, 3H), 7.20 (d, *J* = 6.9 Hz, 2H), 6.52 (s, 1H), 6.37 (s, 1H), 6.17 (s, 2H), 5.93 (d, *J* = 3.7 Hz, 2H), 5.84 (d, *J* = 7.1 Hz, 1H), 5.26–5.10 (m, 1H), 4.41 (d, *J* = 5.1 Hz, 1H), 4.34 (d, *J* = 16.5 Hz, 1H), 4.21 (d, *J* = 16.5 Hz, 1H), 4.08 (t, *J* = 8.3 Hz, 1H), 3.74 (s, 3H), 3.69 (s, 6H), 3.62 (t, *J* = 10.1 Hz, 1H), 2.88–2.73 (m, 1H), 2.57 (dd, *J* = 14.4, 5.1 Hz, 1H); ^13^C NMR (100 MHz, CDCl_3_) *δ* = 173.9, 166.8, 152.7 (2C), 148.7, 147.8, 141.7, 137.4, 134.5, 133.0, 132.5, 132.2 (2C), 129.6 (2C), 129.5 (2C), 128.4, 128.0 (2C), 127.2, 127.1, 110.2, 108.7, 108.3 (2C), 101.8, 68.8, 60.8, 56.3 (2C), 50.6, 43.6, 41.8, 39.5, 36.8; IR *ν*_max_ (KBr): 3437, 2935, 1778, 1535, 1485, 1330, 1234, 933, 748, 609 cm^−^^1^; HRMS (ESITOF) *m/z* calcd. for C_36_H_33_BrN_2_O_9_S, [M + H]^+^ 749.1163, found 749.1166.

*N-((5S,5aS,8aR,9R)-8-Oxo-9-(3,4,5-trimethoxyphenyl)-5,5a,6,8,8a,9-hexahydrofuro[3′,4′:6,7]naphtho[2,3-d][1,3]dioxol-5-yl)-2-phenyl-N′-((4-(trifluoromethyl)phenyl)sulfonyl)acetimidamide* (**4r**), white solid, 69 mg; yield: 93%; m.p.: 245–247 °C, [α]_D_^25^ = −79.2 (c 1.00, CH_2_Cl_2_); ^1^H NMR (400 MHz, CDCl_3_) *δ* = 8.13 (s, 1H), 8.10 (d, *J* = 8.2 Hz, 2H), 7.87 (d, *J* = 8.1 Hz, 2H), 7.41 (d, *J* = 7.4 Hz, 2H), 7.28 (dt, *J* = 13.5, 7.1 Hz, 3H), 6.74 (s, 1H), 6.50 (s, 1H), 6.34 (s, 2H), 5.98 (d, *J* = 5.1 Hz, 2H), 5.48 (dd, *J* = 7.7, 3.4 Hz, 1H), 4.57–4.46 (m, 2H), 4.24 (d, *J* = 14.8 Hz, 1H), 4.13 (dd, *J* = 10.1, 5.3 Hz, 1H), 3.86–3.70 (m, 1H), 3.66 (s, 9H), 3.09 (d, *J* = 4.6 Hz, 2H); ^13^C NMR (100 MHz, CDCl_3_) *δ* = 174.5, 167.8, 153.6 (2C), 149.2, 148.7, 148.2, 138.3, 136.3, 135.9, 133.8, 133.4 (d, *J* = 32.3 Hz, 1C), 130.0 (2C), 129.5 (2C), 129.2, 128.0, 127.9 (2C), 126.8 (q, *J* = 3.8 Hz, 2C), 124.7 (q, *J* = 272.0 Hz, 1C), 110.6, 110.1, 109.5 (2C), 102.5, 69.2, 60.4, 56.4 (2C), 51.4, 44.6, 42.1, 39.7, 37.9; IR *ν*_max_ (KBr): 3267, 2927, 1778, 1539, 1485, 1327, 1234, 1091, 929, 717 cm^−^^1^; HRMS (ESITOF) *m/z* calcd. for C_37_H_33_F_3_N_2_O_9_S, [M + H]^+^ 739.1932, found 739.1936.

*N′-(Methylsulfonyl)-N-((5S,5aS,8aR,9R)-8-oxo-9-(3,4,5-trimethoxyphenyl)-5,5a,6,8,8a,9-hexahydrofuro[3′,4′:6,7]naphtho[2,3-d][1,3]dioxol-5-yl)-2-phenylacetimidamide* (**4s**), white solid, 53 mg; yield: 83%; m.p.: 173–175 °C, [α]_D_^25^ = −55.0 (c 1.00, CH_2_Cl_2_); ^1^H NMR (400 MHz, CDCl_3_) *δ* = 7.39–7.22 (m, 5H), 6.58 (s, 1H), 6.42 (s, 1H), 6.22 (s, 2H), 5.95 (d, *J* = 5.9 Hz, 2H), 5.62 (d, *J* = 6.8 Hz, 1H), 5.22 (dd, *J* = 7.0, 4.5 Hz, 1H), 4.46 (d, *J* = 4.7 Hz, 1H), 4.39 (t, *J* = 8.2 Hz, 1H), 4.36–4.19 (m, 2H), 3.81 (d, *J* = 10.2 Hz, 1H), 3.76 (s, 3H), 3.71 (s, 6H), 3.01 (s, 3H), 2.92 (dtd, *J* = 17.8, 10.7, 8.8, 5.0 Hz, 1H), 2.59 (dd, *J* = 14.4, 4.9 Hz, 1H); ^13^C NMR (100 MHz, CDCl_3_) *δ* = 173.9, 166.8, 152.7 (2C), 148.7, 147.8, 137.4, 134.6, 133.1, 132.5, 129.6 (2C), 129.5 (2C), 128.4, 127.4, 110.2, 108.7, 108.3 (2C), 101.8, 68.9, 60.8, 56.3 (2C), 50.4, 43.6, 43.3, 41.9, 39.4, 37.0; IR *ν*_max_ (KBr): 3452, 2931, 1778, 1585, 1485, 1330, 1234, 1126, 794, 570 cm^−^^1^; HRMS (ESITOF) *m/z* calcd. for C_31_H_32_N_2_O_9_S, [M + H]^+^ 609.1901, found 609.1893.

*N′-(Ethylsulfonyl)-N-((5S,5aS,8aR,9R)-8-oxo-9-(3,4,5-trimethoxyphenyl)-5,5a,6,8,8a,9-hexahydrofuro[3′,4′:6,7]naphtho[2,3-d][1,3]dioxol-5-yl)-2-phenylacetimidamide* (**4t**), white solid, 53 mg; yield: 85%; m.p.: 145–147 °C, [α]_D_^25^ = −60.9 (c 1.00, CH_2_Cl_2_); ^1^H NMR (400 MHz, CDCl_3_) *δ* = 7.39–7.27 (m, 5H), 6.57 (s, 1H), 6.43 (s, 1H), 6.22 (s, 2H), 5.95 (d, *J* = 7.1 Hz, 2H), 5.54 (q, *J* = 9.7, 8.4 Hz, 1H), 5.21 (dd, *J* = 7.2, 4.7 Hz, 1H), 4.46 (d, *J* = 5.0 Hz, 1H), 4.37 (t, *J* = 8.4 Hz, 1H), 4.33–4.24 (m, 2H), 3.83–3.77 (m, 1H), 3.76 (s, 3H), 3.72 (s, 6H), 3.10 (qd, *J* = 7.4, 2.6 Hz, 2H), 2.89 (dq, *J* = 10.6, 3.2 Hz, 1H), 2.60–2.50 (m, 1H), 1.43 (td, *J* = 7.4, 1.9 Hz, 3H); ^13^C NMR (100 MHz, CDCl_3_) *δ* = 173.9, 167.1, 152.7 (2C), 148.7, 147.8, 137.4, 134.6, 133.2, 132.5, 129.6 (2C), 129.5 (2C), 128.4, 127.4, 110.2, 108.7, 108.3 (2C), 101.8, 68.9, 60.8, 56.4 (2C), 50.3, 49.5, 43.6, 41.9, 39.7, 37.0, 8.5; IR *ν*_max_ (KBr): 3402, 2939, 2839, 1778, 1585, 1485, 1388, 1234, 933, 802 cm^−^^1^; HRMS (ESITOF) *m/z* calcd. for C_32_H_34_N_2_O_9_S, [M + H]^+^ 623.2058, found 623.2053.

*N′-(Isobutylsulfonyl)-N-((5S,5aS,8aR,9R)-8-oxo-9-(3,4,5-trimethoxyphenyl)-5,5a,6,8,8a,9-hexahydrofuro[3′,4′:6,7]naphtho[2,3-d][1,3]dioxol-5-yl)-2-phenylacetimidamide* (**4u**), white solid, 50 mg; yield: 77%; m.p.: 138–140 °C, [α]_D_^25^ = −34.3 (c 1.00, CH_2_Cl_2_); ^1^H NMR (400 MHz, CDCl_3_) *δ* = 7.33 (m, 3H), 7.26–7.19 (m, 2H), 6.55 (d, *J* = 12.3 Hz, 1H), 6.42 (d, *J* = 12.4 Hz, 1H), 6.20 (d, *J* = 18.0 Hz, 2H), 5.95 (d, *J* = 5.2 Hz, 2H), 5.50–5.32 (m, 1H), 5.20 (dd, *J* = 7.1, 4.8 Hz, 1H), 4.46 (d, *J* = 5.1 Hz, 1H), 4.43–4.34 (m, 1H), 4.32 (d, *J* = 3.4 Hz, 1H), 3.81 (m, 1H), 3.76 (d, *J* = 7.0 Hz, 3H), 3.71 (d, *J* = 8.5 Hz, 6H), 3.06–2.95 (m, 2H), 2.89 (ddd, *J* = 15.6, 13.1, 9.5 Hz, 1H), 2.58–2.47 (m, 1H), 2.37 (dq, *J* = 13.4, 7.5, 6.6 Hz, 1H), 2.08 (ddt, *J* = 41.5, 13.0, 6.9 Hz, 1H), 1.32–1.08 (m, 6H); ^13^C NMR (100 MHz, CDCl_3_) *δ* = 173.9, 166.7, 152.8 (2C), 148.8, 147.9, 137.5, 134.6, 133.2, 132.5, 129.7 (2C), 128.4, 127.5, 124.7, 122.5, 110.3, 108.7, 108.4, 108.3, 101.8, 68.9, 62.9, 60.8, 56.4 (2C), 50.3, 43.6, 42.0, 39.7, 37.1, 24.8, 22.9, 22.8; IR *ν*_max_ (KBr): 3448, 2931, 1778, 1585, 1535, 1388, 1234, 1126, 933, 725 cm^−^^1^; HRMS (ESITOF) *m/z* calcd. for C_34_H_38_N_2_O_9_S, [M + H]^+^ 651.2371, found 651.2374.

*N′-((((1S,4R)-7,7-Dimethyl-2-oxobicyclo[2.2.1]heptan-1-yl)methyl)sulfonyl)-N-((5S,5aS,8aR,9R)-8-oxo-9-(3,4,5-trimethoxyphenyl)-5,5a,6,8,8a,9-hexahydrofuro[3′,4′:6,7]naphtho[2,3-d][1,3]dioxol-5-yl)-2-phenylacetimidamide* (**4v**), white solid, 50 mg; yield: 67%; m.p.: 143–145 °C, [α]_D_^25^ = −40.7 (c 1.00, CH_2_Cl_2_); ^1^H NMR (400 MHz, CDCl_3_) *δ* = 7.38–7.32 (m, 2H), 7.29 (d, *J* = 6.9 Hz, 3H), 6.58 (s, 1H), 6.42 (d, *J* = 3.3 Hz, 1H), 6.22 (s, 2H), 5.95 (d, *J* = 3.6 Hz, 2H), 5.46 (m, 1H), 5.26 (dd, *J* = 7.3, 4.9 Hz, 1H), 4.51–4.43 (m, 2H), 4.37–4.29 (m, 2H), 3.81 (m, 1H), 3.77 (s, 3H), 3.73 (s, 6H), 3.70 (m, 1H), 3.06–2.88 (m, 2H), 2.76–2.61 (m, 1H), 2.58–2.45 (m, 1H), 2.43–2.32 (m, 1H), 2.18–2.01 (m, 2H), 1.91 (dd, *J* = 18.5, 3.1 Hz, 1H), 1.76 (m, 1H), 1.50–1.39 (m, 1H), 1.14 (d, *J* = 2.6 Hz, 3H), 0.90 (d, *J* = 2.6 Hz, 3H); ^13^C NMR (100 MHz, CDCl_3_) *δ* = 215.7, 174.0, 166.9, 152.7 (2C), 148.7, 147.8, 137.4, 134.5, 133.1, 132.5, 129.8 (2C), 129.7, 129.6, 128.4, 127.6, 110.2, 108.8, 108.3 (2C), 101.8, 69.1, 60.8, 58.5, 56.3 (2C), 51.6, 50.3, 48.3, 43.7, 42.9, 42.7, 42.0, 39.6, 37.0, 27.1, 24.7, 20.0, 19.9; IR *ν*_max_ (KBr): 3441, 2962, 1778, 1743, 1585, 1419, 1234, 1126, 933, 725 cm^−^^1^; HRMS (ESITOF) *m/z* calcd. for C_40_H_44_N_2_O_10_S, [M + H]^+^ 745.2789, found 745.2791.

*N-((5R,5aS,8aR,9R)-9-(4-Hydroxy-3,5-dimethoxyphenyl)-8-oxo-5,5a,6,8,8a,9-hexahydrofuro[3′,4′:6,7]naphtho[2,3-d][1,3]dioxol-5-yl)-2-phenyl-N′-tosylacetimidamide* (**5a**), white solid, 58 mg; yield: 87%; m.p.: 231–233 °C, [α]_D_^25^ = −84.2 (c 1.00, CH_2_Cl_2_); ^1^H NMR (400 MHz, CDCl_3_) *δ* = 7.76 (d, *J* = 7.9 Hz, 2H), 7.33–7.24 (m, 5H), 7.24–7.16 (m, 2H), 6.51 (s, 1H), 6.35 (s, 1H), 6.17 (s, 2H), 5.90 (d, *J* = 10.7 Hz, 2H), 5.79 (d, *J* = 7.1 Hz, 1H), 5.48 (s, 1H), 5.21 (dd, *J* = 7.0, 4.7 Hz, 1H), 4.40–4.31 (m, 2H), 4.18 (d, *J* = 16.5 Hz, 1H), 4.05 (t, *J* = 8.3 Hz, 1H), 3.69 (s, 6H), 3.59 (t, *J* = 10.0 Hz, 1H), 2.78 (dddd, *J* = 15.2, 11.5, 7.5, 4.9 Hz, 1H), 2.55 (dd, *J* = 14.4, 5.0 Hz, 1H), 2.41 (s, 3H); ^13^C NMR (100 MHz, CDCl_3_) *δ* = 174.1, 166.5, 148.6, 147.6, 146.5 (2C), 143.0, 139.9, 134.2, 133.3, 132.7, 130.1, 129.5 (4C), 129.4 (2C), 128.2, 127.3, 126.4 (2C), 110.1, 108.7, 107.9 (2C), 101.7, 68.9, 56.5 (2C), 50.5, 43.4, 41.9, 39.2, 36.7, 21.6; IR *ν*_max_ (KBr): 3502, 3394, 2931, 1778, 1519, 1330, 1230, 1145, 933, 694 cm^−^^1^; HRMS (ESITOF) *m/z* calcd for C_36_H_34_N_2_O_9_S, [M + H]^+^ 671.2058, found 671.2057.

*N′-((4-Chlorophenyl)sulfonyl)-N-((5R,5aS,8aR,9R)-9-(4-hydroxy-3,5-dimethoxyphenyl)-8-oxo-5,5a,6,8,8a,9-hexahydrofuro[3′,4′:6,7]naphtho[2,3-d][1,3]dioxol-5-yl)-2-phenylacetimidamide* (**5b**), white solid, 54 mg; yield: 78%; m.p.: 221–223 °C, [α]_D_^25^ = −87.2 (c 1.00, CH_2_Cl_2_); ^1^H NMR (400 MHz, (CD_3_)_2_CO) *δ* = 8.03–7.90 (m, 1H), 7.89 (dd, *J* = 8.5, 1.0 Hz, 2H), 7.58–7.51 (m, 2H), 7.41 (d, *J* = 7.4 Hz, 2H), 7.34–7.26 (m, 2H), 7.29–7.20 (m, 1H), 7.10 (d, *J* = 1.0 Hz, 1H), 6.70 (s, 1H), 6.48 (s, 1H), 6.32 (s, 2H), 5.99–5.92 (m, 2H), 5.43 (dd, *J* = 7.6, 3.2 Hz, 1H), 4.49 (d, *J* = 14.9 Hz, 2H), 4.22 (d, *J* = 14.8 Hz, 1H), 4.15–4.07 (m, 1H), 3.73 (td, *J* = 8.6, 4.0 Hz, 1H), 3.65 (d, *J* = 1.0 Hz, 6H), 3.05 (d, *J* = 5.3 Hz, 2H); ^13^C NMR (100 MHz, (CD_3_)_2_CO) *δ* = 174.6, 167.4, 149.1, 148.1, 147.9 (2C), 143.9, 138.0, 136.1, 136.0, 134.10, 131.12, 130.0 (2C), 129.7 (2C), 129.5 (2C), 129.2, 128.9 (2C), 127.9, 110.6, 110.0, 109.6 (2C), 102.5, 69.3, 56.6 (2C), 51.3, 44.3, 42.2, 39.5, 37.8; IR *ν*_max_ (KBr): 3510, 3321, 2900, 1766, 1535, 1296, 1226, 933, 798, 759 cm^−^^1^; HRMS (ESITOF) *m/z* calcd. for C_35_H_31_ClN_2_O_9_S, [M + H]^+^ 691.1512, found 691.1509.

*N-((5R,5aS,8aR,9R)-9-(4-Hydroxy-3,5-dimethoxyphenyl)-8-oxo-5,5a,6,8,8a,9-hexahydrofuro[3′,4′:6,7]naphtho[2,3-d][1,3]dioxol-5-yl)-2-phenyl-N′-((4-(trifluoromethyl)phenyl)sulfonyl)acetimidamide* (**5c**), white solid, 60 mg; yield: 82%; m.p.: 169–171 °C, [α]_D_^25^ = −77.9 (c 1.00, CH_2_Cl_2_); ^1^H NMR (400 MHz, (CD_3_)_2_CO) *δ* = 8.10 (d, *J* = 8.2 Hz, 2H), 8.07 (s, 1H), 7.86 (d, *J* = 8.2 Hz, 2H), 7.40 (d, *J* = 7.1 Hz, 2H), 7.33–7.19 (m, 3H), 7.12 (s, 1H), 6.72 (s, 1H), 6.48 (s, 1H), 6.33 (s, 2H), 5.99–5.93 (m, 2H), 5.47 (dd, *J* = 7.6, 3.2 Hz, 1H), 4.57–4.44 (m, 2H), 4.24 (d, *J* = 14.8 Hz, 1H), 4.18–4.06 (m, 1H), 3.76 (td, *J* = 8.7, 3.9 Hz, 1H), 3.65 (s, 6H), 3.07 (q, *J* = 4.7, 3.5 Hz, 2H); ^13^C NMR (100 MHz, (CD_3_)_2_CO) *δ* = 174.6, 167.8, 149.1, 148.6, 148.1, 147.9 (2C), 136.0, 135.9, 134.1, 133.4 (q, *J* = 32.5 Hz, 1C), 131.1, 130.0 (2C), 129.5 (2C), 129.1, 128.0, 127.9 (2C), 126.76 (q, *J* = 3.8 Hz, 2C), 124.7 (d, *J* = 271.9 Hz, 1C), 110.6, 110.0, 109.6 (2C), 102.5, 69.2, 56.6 (2C), 51.4, 44.3, 42.2, 39.6, 37.8; IR *ν*_max_ (KBr): 3541, 3305, 2939, 1766, 1539, 1400, 1327, 1141, 937, 721 cm^−1^; HRMS (ESITOF) *m/z* calcd. for C_36_H_31_F_3_N_2_O_9_S, [M + H]^+^ 725.1775, found 725.1784.

*4-(-2-(((5R,5aS,8aR,9R)-9-(4-Hydroxy-3,5-dimethoxy phenyl)-8-oxo-5,5a,6,8,8a,9-hexahydrofuro[3′,4′:6,7]naphtho[2,3-d][1,3]dioxol-5-yl)amino)-2-(tosylimino)ethyl)benzoic acid* (**5d**), white solid, 40 mg; yield: 55%; m.p.: 245–247 °C, [α]_D_^25^ = −56.0 (c 1.00, CH_2_Cl_2_); ^1^H NMR (400 MHz, (CD_3_)_2_CO) *δ* = 7.99 (d, *J* = 7.5 Hz, 1H), 7.93 (d, *J* = 7.9 Hz, 2H), 7.79 (d, *J* = 7.9 Hz, 2H), 7.52 (d, *J* = 7.9 Hz, 2H), 7.33 (d, *J* = 8.0 Hz, 2H), 6.76 (s, 1H), 6.48 (s, 1H), 6.33 (s, 2H), 5.97 (d, *J* = 4.7 Hz, 2H), 5.44 (dd, *J* = 7.6, 3.3 Hz, 1H), 4.63 (d, *J* = 15.0 Hz, 1H), 4.48 (d, *J* = 3.9 Hz, 1H), 4.26 (d, *J* = 15.0 Hz, 1H), 4.12–4.03 (m, 1H), 3.79–3.67 (m, 2H), 3.66 (s, 6H), 3.13–2.98 (m, 2H), 2.39 (s, 3H); ^13^C NMR (100 MHz, (CD_3_)_2_CO) *δ* = 174.7, 167.5, 166.2, 149.1, 148.1, 147.9 (2C), 143.1, 142.2, 141.5, 136.1, 134.1, 131.2, 130.7 (2C), 130.3, 130.1 (2C), 130.0 (2C), 129.4, 127.2 (2C), 110.6, 110.0, 109.6 (2C), 102.5, 69.3, 56.6 (2C), 51.2, 44.4, 42.2, 39.4, 37.9, 21.4; IR *ν*_max_ (KBr): 3541, 3437, 2900, 1766, 1697, 1539, 1427, 1226, 929, 690 cm^−1^; HRMS (ESITOF) *m/z* calcd. for C_37_H_34_N_2_O_11_S, [M + H]^+^ 715.1956, found 715.1955.

*2-(4-(Dimethylamino)phenyl)-N-((5R,5aS,8aR,9R)-9-(4-hydroxy-3,5-dimethoxyphenyl)-8-oxo-5,5a,6,8,8a,9-hexahydrofuro[3′,4′:6,7]naphtho[2,3-d][1,3]dioxol-5-yl)-N′-tosylacetimidamide* (**5e**), white solid, 48 mg; yield: 67%; m.p.: 244–146 °C, [α]_D_^25^ = −74.2 (c 1.00, CH_2_Cl_2_); ^1^H NMR (400 MHz, CDCl_3_) δ = 7.84 (d, J = 7.8 Hz, 2H), 7.36–7.23 (m, 2H), 7.00 (d, J = 8.1 Hz, 2H), 6.69–6.52 (m, 3H), 6.40 (s, 1H), 6.19 (s, 2H), 5.94 (d, J = 6.2 Hz, 2H), 5.57–5.37 (m, 2H), 5.24 (d, J = 6.0 Hz, 1H), 4.48–4.37 (m, 1H), 4.20 (s, 2H), 4.11 (t, J = 8.7 Hz, 1H), 3.73 (s, 6H), 3.61 (t, J = 10.3 Hz, 1H), 2.93 (s, 6H), 2.78 (d, J = 13.3 Hz, 1H), 2.45 (m, 1H), 2.42 (s, 3H); ^13^C NMR (100 MHz, CDCl_3_) δ = 174.0, 167.9, 150.3, 148.7, 147.7, 146.6 (2C), 143.0, 140.1, 134.3, 132.7, 130.6 (2C), 130.0, 129.6 (2C), 127.5, 126.6 (2C), 119.2, 113.2 (2C), 110.2, 108.8, 107.9 (2C), 101.7, 69.1, 56.6 (2C), 50.3, 43.5, 42.0, 40.4 (2C), 38.6, 36.8, 21.6; IR ν_max_ (KBr): 3541, 3321, 2897, 1766, 1523, 1481, 1280, 1226, 937, 682 cm^−1^; HRMS (ESITOF) *m/z* calcd. for C_38_H_39_N_3_O_9_S, [M + H]^+^ 714.2480, found 714.2471.

*N-((5R,5aS,8aR,9R)-9-(4-hydroxy-3,5-dimethoxyphenyl)-8-oxo-5,5a,6,8,8a,9-hexahydrofuro[3′,4′:6,7]naphtho[2,3-d][1,3]dioxol-5-yl)-2-(3-hydroxyphenyl)-N′-tosylacetimidamide* (**5f**), white solid, 47 mg; yield: 68%; m.p.: 175–177 °C, [α]_D_^25^ = −76.6 (c 1.00, CH_2_Cl_2_); ^1^H NMR (400 MHz, (CD_3_)_2_CO) *δ* = 8.45 (s, 1H), 7.80 (d, *J* = 8.0 Hz, 2H), 7.70 (d, *J* = 7.5 Hz, 1H), 7.33 (d, *J* = 8.0 Hz, 2H), 7.15–7.05 (m, 2H), 6.91 (d, *J* = 2.4 Hz, 1H), 6.87–6.82 (m, 1H), 6.72 (dd, *J* = 8.1, 2.5 Hz, 1H), 6.67 (s, 1H), 6.45 (s, 1H), 6.32 (s, 2H), 5.94 (d, *J* = 1.0 Hz, 2H), 5.40 (dd, *J* = 7.6, 3.3 Hz, 1H), 4.46–4.38 (m, 2H), 4.12 (d, *J* = 14.9 Hz, 1H), 4.09–4.02 (m, 1H), 3.71 (dtd, *J* = 9.5, 5.5, 4.4, 1.7 Hz, 1H), 3.64 (d, *J* = 1.0 Hz, 6H), 3.03 (m, 2H), 2.38 (s, 3H); ^13^C NMR (100 MHz, (CD_3_)_2_CO) *δ* = 174.7, 167.2, 158.5, 149.0, 148.0, 147.9 (2C), 143.0, 142.3, 137.5, 135.9, 134.0, 131.2, 130.4, 130.1 (2C), 129.3, 127.1 (2C), 121.0, 117.1, 114.9, 110.5, 109.9, 109.5 (2C), 102.4, 69.4, 56.6 (2C), 51.0, 44.3, 42.2, 39.2, 37.8, 21.4; IR *ν*_max_ (KBr): 3616, 3433, 2943, 1774, 1519, 1388, 1280, 1230, 933, 690 cm^−1^; HRMS (ESITOF) *m/z* calcd. for C_36_H_34_N_2_O_10_S, [M + H]^+^ 687.2007, found 687.2003.

### 2.3. Biological Assay

The A-549 cells and MRC-5 cells were obtained from the American Type Culture Collection and cultured in an environment of 5% CO_2_ at 37 °C in RPMI-1640 medium supplemented with 10% fetal bovine serum (FBS). Lung (A-549) human cancer cells were seeded in 96-well plates at a density of 3000 cells/well in normoxia for 12 h. Then, measures of 100 μL drug-containing medium, with a series of concentrations, were dispensed into the wells to attain the final concentration as 100, 80, 20, 10, 5, and 2 μM. After 48 h incubated under normoxia or hypoxia, 20 μL MTT solution (Beyotime Biotechnology, Nantong, China, 5 mg/mL MTT dissolved in PBS) was added. Then, following incubation for another 4 h, the medium was discarded, followed by the addition of 200 μL DMSO. The absorbance was measured at 570 nm with a microplate reader. Experiments were conducted in triplicate. The IC_50_ values are the average of at least three independent experiments.

### 2.4. Molecular Docking

The crystal structure of topoisomerase-II (PDB ID:4G0U) was downloaded from the PDB database. In the calculation process, the preliminary processing of the target (protein structure modification, deletion of invalid residues and original ligand) was completed by DS2019. CDOCKER software was used for docking, and the receptor was set as rigid, while the compound was set as flexible. Glu461, Gly462, Asp463, Arg487, and Gly488 were selected as docking sites for docking. Visualization and interaction force analysis were performed using pymol2.3 and ligplot2.2.

### 2.5. Molecular Dynamics

GROMACS 2019.4 ran on a high-performance Linux cluster to determine the behavior of the PSAH with topoisomerase-II within 100 ns. We used the Bio2byte Web server (https://www.bio2byte.be/ accessed on 12 July 2021) to generate topology files for the PSAH. From the docking study, the complex with the most drug-forming activity and the best docking status was selected as the input file of MD simulation. The force field uses amber99sb-ildn.ff. The complex of the TIP3P water model was surrounded by the dodecahedral-shaped water tank. In order to neutralize the net charge of the system, Na and Cl counter ions were replaced by water molecules. The steepest descent algorithm with a tolerance of 1000 kJ/mol/nm was used to minimize the energy of the system. The cutoff value of van der Waals was 12 Å, and periodic boundary conditions were specified in all directions. After convergence, the NVT ensemble MD simulation was within 100 ps, and then the system passed through NPT within 100 ps under periodic boundary conditions. Berendsen constant pressure and a thermostat were used to maintain the temperature and pressure at 300 K and 1 bar for coupling times of τ T = 0.1 ps and τ p = 2 ps. The particle grid Ewald (PME) method was used to calculate the long-range electrostatic interaction. The LINCS algorithm was used to limit the key length. For PSAH, the MD simulation run of 100 ns was repeated twice at a constant temperature and pressure, and the average value of the results was reported.

## 3. Results and Discussion

### 3.1. Chemistry

The key reaction in the synthesis of this class of compounds was the copper-catalyzed sulfonyl azide–alkyne cycloaddition/ring cleavage (CuAAC/ring-opening reaction) [44], which has been applied to synthesize numerous oxygen-containing and nitrogen-containing heterocyclic compounds [45]. As show in Figure 2, podophyllotoxins **1a** and **1b** were easily prepared by the substrate podophyllotoxin nucleophilic substitution and reduction with NaN_3_. Initially, podophyllotoxin (**1a** or **1b**) was dissolved in MeCN; then, corresponding terminal alkynes (**2a**–**2m**, see Appendix A) and sulfonyl azides (**3a**–**3j**, see Appendix A) were added with stirring. The reaction mixture was stirred for about 12 h to afford podophyllotoxin-*N*-sulfonyl amidine hybrids (**4a**–**4v** and **5a**–**5f**) in good–excellent yields (Table 1). This reaction appears quite flexible and offers an easy capacity to generate a large scale PSAH with mild condition, operability, and highly atom economical. The chemical structures of all synthesized compounds were determined by ^1^H NMR, ^13^C NMR, IR, and HRMS.

### 3.2. Interpretation of Spectral Data

For compound **4a** (Figure 3), the ^1^H NMR chemical shifts of the characteristic functional groups are as follows: *δ* (C^8^-CH_2_) = 5.95 (d, *J* = 3.0 Hz, 2H); *δ* (C^7′^,C^9′^-OCH_3_) = 3.71 (s, 6H); *δ* (C^8′^-OCH_3_) = 3.77 (s, 3H); *δ* (C^24^-CH_3_ of Ts group) = 2.43 (s, 3H). The ^13^C NMR chemical shifts of the characteristic functional groups are as follows: *δ* (C^13^: C=O) = 173.9; *δ* (C^14^: C=N) = 166.5; *δ* (C^3′^+C^5′^: Aryl (C)-O) = 152.8 (2C); *δ* (C^8^: O-CH_2_-O) = 101.9; *δ* (C^12^: -CH_2_-O) = 69.0; *δ* (C^8′^: -O-CH_3_) = 60.9; *δ* (C^7′^+C^9′^: -O-CH_3_) = 56.4 (2C); *δ* (C^24^: CH_3_ of Ts group) = 21.7 (C^24^). The detailed explanation of spectral data for **4a** is as follows: **^1^**H NMR (400 MHz, CDCl_3_) *δ* = 7.82 (dd, *J* = 15.7, 7.9 Hz, 2H, C^21^, C^21′^-H), 7.33 (dd, *J* = 11.9, 7.4 Hz, 5H, C^22^, C^22′^, C^18^, C^18′^, C^19^-H), 7.21 (d, *J* = 7.2 Hz, 2H, C^17^, C^17′^-H), 6.52 (s, 1H, C^6^-H), 6.41 (s, 1H, C^10^-H), 6.17 (s, 2H, C^2′^, C^6′^-H), 5.95 (d, *J* = 3.0 Hz, 2H, C^8^-H), 5.27 (d, *J* = 6.9 Hz, 1H, C^4^-H), 5.19 (t, *J* = 5.7 Hz, 1H, C^1^-H), 4.42 (d, *J* = 5.1 Hz, 1H, C^12^-H), 4.41–4.27 (m, 2H, C^15^-H), 4.10 (t, *J* = 8.3 Hz, 1H, C^12^-H), 3.77 (s, 3H, C^8′^-H), 3.71 (s, 6H, C^7′^,C^9′^-H), 3.61 (t, *J* = 10.0 Hz, 1H, N-H), 2.86–2.68 (m, 1H, C^2^-H), 2.45 (m, 1H, C^3^-H), 2.43 (s, 3H, C^24^-H); **^13^**C NMR (100 MHz, CDCl_3_) *δ* = 173.9 (C^13^), 166.5 (C^14^), 152.8 (2C, C^3′^+C^5′^), 148.8 (C^7^), 147.9 (C^9^), 143.2 (C^23^), 140.0 (C^20^), 137.5 (C^4′^), 134.5 (C^16^), 133.0 (C^1′^), 132.6 (C^11^), 129.9 (C^5^), 129.8 (2C, C^22^+C^22′^), 129.7 (2C, C^18^+C^18′^), 128.6 (C^21^), 127.3 (C^21′^), 126.6 (2C, C^17^+C^17′^), 126.5 (C^19^), 110.3 (C^10^), 108.7 (C^6^), 108.4 (2C, C^2′^+C^6′^), 101.9 (C^8^), 69.0 (C^12^), 60.9 (C^8′^), 56.4 (2C, C^7′^+C^9′^), 50.6 (C^4^), 43.7 (C^2^), 42.0 (C^1^), 39.5 (C^3^), 36.9 (C^15^), 21.7 (C^24^). The compound **4e**, especially, contains two isomers. The reason for the formation is that the hydrogen atom of the hydroxyl group forms a hydrogen bond with the oxygen atom of the lactone ring carbonyl group, which are close to each other in space, resulting in rotational isomerism.

### 3.3. Biological Study

All the newly synthesized PSAH derivatives were evaluated for their in vitro inhibitory activity toward A-549 cells using a MTT assay. The results expressed as half-maximal inhibitory concentration (IC_50_) values are presented in Table 1. The IC_50_ values are the average of at least three independent experiments. Agreeably, as shown in Table 1, most PSAH, whether they have an electron-donating or electron-withdrawing nature, have shown moderate–good anticancer activity in this investigation. When containing the -Me (**4b**), -OMe (**4d**), -Cl (**4f**, **4g**), -Br (**4h**, **4q**), -CO_2_H (**4j**, **5d**), 1-Indole (**4m**), or 5-hydrindenyl (**4o**) functional groups, the PSAH product exhibited slight significant anticancer activity with high IC_50_ values exceeded 100 μM, while the -Me_2_N (**4c**, **5e**), -OH (**4e**), 2-thienyl (**4l**), or aliphatic group and the substituted R^2^ (**4s**–**4u**) group gave positive apoptotic activity with IC_50_ values ranging from 12.7 to 92.0 μM. The compounds with the -CF_3_ (**4r**, **5c**) or 10-camphor (**4v**) group have shown the most promising apoptotic activity among the other substituents (for **4r**, 5.21 μM, **4v**, 2.44 μM, and **5c**, 1.65 μM), which exhibited more potent activity than standard drug etoposide (12 ± 0.12 μM against A-549 [46]). Relative to other functional groups, the strong electron-sucking effect of the -CF_3_ group may have increased the attraction between the protein and the **PSAH [47]**. On the whole, modifying R^1^ or R^3^ is more effective than R^2^. Besides, we tested the healthy cell line (MRC-5 cells) of the most active compounds—**5c**, **4v**, and **4r**—and found that all the IC_50_ values exceeded 80 μM, which shows their low toxicity to MRC-5 cells.

### 3.4. Molecular Docking

In order to check the affinity of topoisomerase-II and PSAH **5c**, docking results are shown in Figure 1, where the docking score of both is −8.9 kcal/mol. From Figure 4A, we are able to see the binding mode and conformation of **5c** at the target. The green residue (ASN-509) has hydrogen bond interaction with **5c**. The detailed interaction force can be seen from Figure 4B. Asn-509 has hydrogen bond interaction with ligand with bond lengths of 3.23 and 2.87. Pro-439, ARG-487, PHE484, ASP463, and LYS-440 formed hydrophobic interactions. The positive control compound Etoposide (−8.3 kcal/mol), which forms hydrogen bond interactions with residue His-759, and forms hydrophobic interactions with Gln-461, Ser-464, and Ala-465. After comparison, it can be found that the binding effect of the selected compound **5c** is better than that of the positive compound.

### 3.5. Molecular Dynamics

In order to test the stability of topoisomerase-II binding with PSAH **5c**, a 100 ns molecular dynamics simulation was carried out, and its output was analyzed as follows. This simulation result is supplemented in the following material (Figure 4). In the 100 ns simulation, the complex converged to an equilibrium state at 27 ns and was able to remain stable until the end of the simulation. The RMSD value was 0.6 nm. The root mean square fluctuation (RMSF) of each residual basis in Figure 5 was analyzed in detail to determine residual fluctuation and flexibility during the whole simulation. The overall root mean square density of the compound was very low, between 0.1 nm and 0.6 nm. Finally, from RMSD and RMSF, the system is stable. *N*-sulfonyl amidine groups did not affect the molecular docking and stability, but enhanced the antiproliferative activity.

## 4. Conclusions

All the new podophyllotoxin-*N*-sulfonyl amidine hybrids (PSAH) were synthesized and evaluated for their antiproliferative activity against human lung (A-549) cancer cell lines. The experimental results indicated that most of the synthesized compounds showed moderate–potent antiproliferative activity, and compound **5c** exhibited excellent antiproliferative activity against human lung (A-549) cancer cell lines of 1.65 μM. Molecular docking and molecular dynamics analyses revealed that the promising antibacterial efficacy of PSAH can be attributed to the substitution of chlorine on the benzene and the modification of the coumarin core. The combination effect of topoisomerase-II and **5c** is tight. PSAH is closely associated with topoisomerase-II in a stable system. The results of the present work showed that podophyllotoxin-*N*-sulfonyl amidine hybrids have potential to be antineoplastic drugs and are worthy of further study.

## Data Availability

The data is contained within the article.

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
