# Peer review of "Design and Synthesis of Novel Podophyllotoxins Hybrids and the Effects of Different Functional Groups on Cytotoxicity"

_molecules, 2021, doi:10.3390/molecules27010220_

Round 1

Reviewer 1 Report

The article written by Zhong-Tao Yang et al. reports studies on a new series of podophyllotoxin-N-sulfonyl amidine hybrids.

It is well known that podophyllotoxin derivatives exhibit high anticancer activity. Therefore, it is not clear to me why have the Authors examined activity of the new compounds solely for one cancel cell line (A549). In my opinion, this study can be considered as preliminary, and activity against other cancer cell lines should be examined.

The toxicity of the obtained compounds on the healthy cell line was not tested either. Therefore, it is unknown whether the most active compound would be useful as drug. Toxicity study should be included.

The reported products are optically active. Where they obtained as enantiomerically pure? Please measure the optical rotation for each compound.

The authors wrote that “The strong electron-sucking effect of –CF3 group maybe increase the attraction between protein and PSAH” [47]. Is this effect confirmed by the molecular modeling carried out by the authors?

Please calculate the docking score for podophyllotoxin in kcal / mol and compare with the obtained IC50 values.

In the Conclusions section, the authors wrote that the activity of 5c was 16.5 microM instead of 1.65 microM. Please, correct.

Author Response

Reviewer: 1

1. It is well known that podophyllotoxin derivatives exhibit high anticancer activity. Therefore, it is not clear to me why have the Authors examined activity of the new compounds solely for one cancel cell line (A549). In my opinion, this study can be considered as preliminary, and activity against other cancer cell lines should be examined.

Reply: Thanks for the Reviewer’s comments. Because one of the ingredients (NaN3) is a chemical substance controlled by the Ministry of Public Security, we have difficulty synthesizing enough products for testing, so far only enough for examining A-549 cell lines. Our future studies will consider the source of ingredients and test more cell lines.

2. The toxicity of the obtained compounds on the healthy cell line was not tested either. Therefore, it is unknown whether the most active compound would be useful as drug. Toxicity study should be included.

Reply: Thanks for the Reviewer’s comments. We have tested the healthy cell line of the most active compound 5c, 4v, 4r, and added the description in the section 3.3.

3. The reported products are optically active. Where they obtained as enantiomerically pure? Please measure the optical rotation for each compound.

Reply: Thanks for the Reviewer’s comments. We have tested the optical rotation for each compound, and added the description in compound characterizations and instrument information in the section 2.2.

4. The authors wrote that “The strong electron-sucking effect of –CF3 group maybe increase the attraction between protein and PSAH” [47]. Is this effect confirmed by the molecular modeling carried out by the authors?

Reply: We are sorry to make this unclear concept. The conclusions inferred from ref. 47 and our experiments. Thus, the sentences change to “Relative to other functional groups, the strong electron-sucking effect of –CF3 group maybe increase the attraction between protein and PSAH.”

5. Please calculate the docking score for podophyllotoxin in kcal / mol and compare with the obtained IC50 values.

Reply: Thanks. We accepted the suggestion and calculated the docking score for Etoposide instead of podophyllotoxin. We focused on the small-molecule 5c as the most promising candidate inhibitor with a docking score -8.9 kcal/mol, compare with Etoposide -8.3 kcal/mol. We added this calculation in the section 3.4.

6. In the Conclusions section, the authors wrote that the activity of 5c was 16.5 microM instead of 1.65 microM. Please, correct.

Reply: We are sorry to make this mistake by a clerical error. It should be 1.65 and it has been corrected in the revised manuscript.

Reviewer 2 Report

The manuscript entitled “Design, synthesis of novel podophyllotoxins hybrids and the effects of different functional groups on cytotoxicity” by Zhong-Tao Yang et al. described the synthesis of a new series of podophyllotoxin-N-sulfonyl amidine hybrids using CuAAC/ring-opening procedure and the study of the influence of functional groups on antiproliferative activity. The manuscript may be of general interest to the researchers of this field, but the manuscript lacks some information that the author should consider and incorporate in the present form of the manuscript. Here are a few concerns that need to be addressed in the present form of the manuscript.

  1. The background addressed in a broad context and highlight the purpose of the study should be added in the abstract.
  2. The keyword "structure-activity correlation" should be added in keywords
  3. Scheme 2 should be replaced from part “Materials and Methods” to “Results and Discussion, 3.1. Chemistry”.
  4. Interpretation of spectral data for synthesized compounds, especially the formation of two isomers for compound 4e, should be discussed in the text “Results and Discussion, 3.1. Chemistry”.
  5. Compound characterizations should be replaced from “Appendix A” to “2.2. General procedure for the synthesis of PSAH” after general procedure.
  6. In Conclusions the authors conclude that molecular docking and molecular dynamics analysis revealed that PSAH the promising antibacterial efficacy. While in the next sentence, as in the article, the authors talk about antitumor activity. The authors should explain the conclusion about potential antibacterial activity.

Author Response

1.The background addressed in a broad context and highlight the purpose of the study should be added in the abstract.

Reply: We accepted the suggestion and added the background and purpose in the abstract: “Development of novel anticancer therapeutic candidates is one of the key challenge in medicinal chemistry. Podophyllotoxin and its derivatives as the potent cytotoxic agent have been the centre of extensive chemical amendment and pharmacological investigation.”

2. The keyword "structure-activity correlation" should be added in keywords

Reply: We accepted the suggestion and added the relevant content in the revised manuscript.

3. Scheme 2 should be replaced from part “Materials and Methods” to “Results and Discussion, 3.1. Chemistry”.

Reply: We accepted the suggestion and it has moved the Scheme 2 to the section 3.1 in the revised manuscript.

4. Interpretation of spectral data for synthesized compounds, especially the formation of two isomers for compound 4e, should be discussed in the text “Results and Discussion, 3.1. Chemistry”.

Reply: We accepted the suggestion and added the interpretation as the section 3.2 in the revised manuscript, and adjust the serial number behind.

5. Compound characterizations should be replaced from “Appendix A” to “2.2. General procedure for the synthesis of PSAH” after general procedure.

Reply: We accepted the suggestion and moved the compound characterizations to the section 2.2 in the revised manuscript.

6. In Conclusions the authors conclude that molecular docking and molecular dynamics analysis revealed that PSAH the promising antibacterial efficacy. While in the next sentence, as in the article, the authors talk about antitumor activity. The authors should explain the conclusion about potential antibacterial activity.

Reply: We are sorry to make this mistake by a clerical error. It should be antitumor activity and it has been corrected in the revised manuscript.

Other

The full text was checked carefully and some words and grammars were corrected.

Round 2

Reviewer 1 Report

The authors have answered all my questions and comments. I recommend acceptance of the manuscript in its current form.